# Low Vitamin C Status in Patients with Cancer Is Associated with Patient and Tumor Characteristics

**DOI:** 10.3390/nu12082338

**Published:** 2020-08-05

**Authors:** Rebecca White, Maria Nonis, John F. Pearson, Eleanor Burgess, Helen R. Morrin, Juliet M. Pullar, Emma Spencer, Margreet C. M. Vissers, Bridget A. Robinson, Gabi U. Dachs

**Affiliations:** 1Mackenzie Cancer Research Group, Department of Pathology and Biomedical Science, University of Otago Christchurch, Christchurch 8011, New Zealand; whire874@student.otago.ac.nz (R.W.); nonpa990@student.otago.ac.nz (M.N.); eleanor.burgess@postgrad.otago.ac.nz (E.B.); helen.morrin@otago.ac.nz (H.R.M.); bridget.robinson@cdhb.health.nz (B.A.R.); 2Biostatistics and Computational Biology Unit, University of Otago Christchurch, Christchurch 8011, New Zealand; john.pearson@otago.ac.nz; 3Cancer Society Tissue Bank, University of Otago Christchurch, Christchurch 8011, New Zealand; 4Centre for Free Radical Research, Department of Pathology and Biomedical Science, University of Otago Christchurch, Christchurch 8011, New Zealand; juliet.pullar@otago.ac.nz (J.M.P.); margreet.vissers@otago.ac.nz (M.C.M.V.); 5Nutrition in Medicine Research Group, Department of Pathology and Biomedical Science, University of Otago Christchurch, Christchurch 8011, New Zealand; emma.spencer@otago.ac.nz; 6Canterbury Regional Cancer and Hematology Service, Canterbury District Health Board, and Department of Medicine, University of Otago Christchurch, Christchurch 8011, New Zealand

**Keywords:** ascorbate, breast cancer, colorectal cancer, chemotherapy, immunotherapy, surgery, exercise

## Abstract

Vitamin C (ascorbate) acts as an antioxidant and enzyme cofactor, and plays a vital role in human health. Vitamin C status can be affected by illness, with low levels being associated with disease due to accelerated turnover. However, robust data on the ascorbate status of patients with cancer are sparse. This study aimed to accurately measure ascorbate concentrations in plasma from patients with cancer, and determine associations with patient or tumor characteristics. We recruited 150 fasting patients with cancer (of 199 total recruited) from two cohorts, either prior to cancer surgery or during cancer chemo- or immunotherapy. A significant number of patients with cancer had inadequate plasma ascorbate concentrations. Low plasma status was more prevalent in patients undergoing cancer therapy. Ascorbate status was higher in women than in men, and exercising patients had higher levels than sedentary patients. Our study may prompt increased vigilance of ascorbate status in cancer patients.

## 1. Introduction

The role of vitamin C (ascorbate) as an anti-cancer agent has long been debated [1,2]. There is interest in the use of ascorbate by patients with cancer, either through dietary supplementation or pharmacological infusion dosing [3,4], and isolated reports of clinical benefit have been published (reviewed in [5]). However, there are limited data available on the ascorbate status of this patient group. As humans rely solely on dietary intake for ascorbate, due to an inability to synthesize the vitamin, guidelines for intake to prevent the severe ascorbate deficiency disease scurvy have been developed [6]. Recommended daily intake (RDI) guidelines vary between countries, with Australasian standards (RDI of 45 mg/day for both males and females [7]) being among the lowest worldwide. In comparison, the United States guidelines recommend a daily intake of 90 mg for males and 75 mg for females over the age of 19 [8], and the German Nutrition Society has calculated that healthy men required 110 mg and women 95 mg ascorbate per day [9].

The physiological effects of ascorbate are achieved through its ability to donate electrons, either as an antioxidant or as a cofactor for metal-containing enzymes [10,11]. Particularly pertinent to cancer are the 2-oxoglutarate-dependent dioxygenases, a superfamily of enzymes, for which ascorbate is an essential cofactor [11]. These ascorbate-reliant enzymes have roles in wound healing via the synthesis and stabilization of collagen, and in mood and vitality regulation via synthesis of carnitine, neurotransmitters and peptide hormones [11]. This superfamily also includes DNA and histone demethylases, that modify cancer stem cell phenotype, and hydroxylases that regulate the hypoxic response via transcription factor activation, which affects cancer progression [2,11]. Retrospective analysis of human tissue samples revealed that increased tumor ascorbate levels were associated with reduced transcription factor activity (specifically hypoxia-inducible factors) in endometrial, colorectal, thyroid, papillary renal cell and breast cancer [12,13,14,15,16]. Increased tumor ascorbate levels were associated with extended disease-free survival in colorectal cancer patients [13] and improved disease-specific survival in breast cancer patients [16]. These data indicate that optimal levels of ascorbate may be clinically important in cancer.

Despite this interest in ascorbate for cancer, there are limited data on the ascorbate status of patients with cancer, with many studies involving small sample sizes (median *n* = 50, with 27/31 studies having fewer than 100 patients) [17,18,19,20,21,22]. In addition, most used the colorimetric assay (21/24 studies) (e.g., [17,18,19]), which can be prone to interference by other reducing substances present in plasma [23]. Only three studies used high-performance liquid chromatography (HPLC, the gold standard for measuring ascorbate), all three containing fewer than 70 patients [20,21,22]. Most studies (20/31) reported inadequate plasma ascorbate levels in patients with cancer e.g., [17,18,19,20,21,22].

The aim of our study was therefore to measure the concentration of ascorbate in plasma using HPLC in a cohort of outpatients with cancer. Plasma levels of ascorbate are indicative of whole body status and correlate with tissue levels, with plasma saturation being reflective of general tissue saturation [6,24]. Of interest were newly diagnosed patients prior to the planned surgical resection of their cancer (largely treatment naïve), as well as patients undergoing cancer therapy (chemotherapy or immunotherapy). We hypothesized that certain lifestyle factors, patient characteristics or tumor characteristics would predict patients at risk of deficiency or hypovitaminosis C. We also hypothesized that therapy patients may be at particular risk of ascorbate depletion due to reduced intake or increased requirements.

## 2. Materials and Methods

### 2.1. Human Ethics and Patient Consent

Ethics approval for this study was obtained from the New Zealand Health and Disability Ethics Committee (18/STH/223). Patients were recruited between November 2018 and June 2019 at Christchurch Hospital, New Zealand, to either a pre-surgical cohort or a therapy cohort. Patients gave informed consent for a blood draw of 5 mL via venipuncture, for administration of a health questionnaire and for collection of related medical data. Patients declared their ethnicity using the New Zealand census question and were offered the option of disposal of samples by karakia (blessing).

### 2.2. Eligibility Criteria

Participants were aged over 18 with a confirmed diagnosis of cancer and able to provide informed consent. Participants were not required to be fasting prior to blood draw, but their food and drink consumption, and any ascorbate supplementation, on the day of blood draw was recorded.

### 2.3. Study Populations

Patients in the pre-surgical cohort had confirmed cancer that was to be surgically resected and were thus recruited from the surgical pre-admission clinics in Christchurch Hospital. Patients in the therapy cohort were recruited from the chemotherapy day ward at Christchurch Hospital and were partway through their treatment schedule for cancer having received at least one prior treatment (chemotherapy or immunotherapy) on the day of recruitment. All participants completed a brief health questionnaire to collect demographic variables (details below). Medical records provided clinicopathological information (cancer type, Tumor Node Metastasis (TNM) stage) and treatment details (type of chemotherapy or immunotherapy).

### 2.4. Blood Sample Collection and Processing

Peripheral blood was collected into 4 mL Ethylene-diamine-tetra-acetic acid (EDTA) vacutainer tubes (Becton Dickinson, Auckland, New Zealand), immediately placed on ice and processed within 60 min. Samples were centrifuged at 4 °C to pellet cells, and plasma was collected for the extraction of ascorbate. Plasma was mixed at a ratio of 1:1 with ice-cold 0.54 M perchloric acid (PCA) solution containing 100 μM of the metal chelator di-ethylenetriamine penta-acetic acid (DTPA) to precipitate protein (chemicals from Sigma-Aldrich, St Louis, MO, USA). Cleared plasma was stored at −80 °C until analysis.

### 2.5. Ascorbate Measurement in Plasma

Total ascorbate content of plasma was analyzed using reverse-phase HPLC with electrochemical detection, as described previously [25]. Samples were reduced with 10 mg/mL tris-(2-carboxyethyl)-phosphine (TCEP) at a 1:10 ratio for 3 h at 4 °C, and diluted 1:1 in 77 mM perchloric acid containing DTPA (100 µM). Samples were separated on a Synergi 4 µ Hydro-RP 80A 150 × 4.6 mm column (Phenomenex NZ Ltd., Auckland, New Zealand) using a Dionex Ultimate 3000 HPLC unit with a refrigerated autosampler and an Ultimate 3000 ECD−3000RS electrochemical detector and a Model 6011RS coulometric analytical cell (+200 mV electrode potential). The mobile phase comprised 80 mM sodium acetate buffer, pH 4.8, containing DTPA (0.54 mM) and freshly added paired-ion reagent *n*-octylamine (1 µM), delivered at a flow rate of 1.2 mL/min. A standard curve of sodium-L-ascorbate (1.25–40 μM), standardized spectrophotometrically, was freshly prepared for each HPLC run in 77 mM perchloric acid containing 100 µmol/L DTPA. All chemicals were from Sigma-Aldrich (St Louis, MO, USA). Plasma ascorbate concentrations are expressed as μM and patients were classified according to international clinical guidelines as follows: deficient < 11 μM, marginal 11 ≤ 23 μM, inadequate 23–50 μM or adequate >50 μM [25]; >70 μM is deemed saturating.

### 2.6. Health Data and Dietary Intake Assessment

Health questionnaires provided age, gender, height and weight; height and weight were used to calculate BMI (kg/m^2^). Frequency of physical activity was recorded to be <60 min/week, 60–150 min/week or >150 min/week, according to Ministry of Health NZ guidelines [26]). Smoking habits were recorded as: never smoked; ex-smoker or current smoker. Patient’s diet on the day prior to sampling (24-hour dietary recall) was used to estimate ascorbate intake using the US Department of Agriculture database [27]. In addition, reported intake on the morning of blood draw enabled patients to be separated into fasting and non-fasting groups. Those patients who had consumed less than 45 mg ascorbate and who had not taken any ascorbate supplements on the day of blood draw, were defined as being fasting, with respect to ascorbate intake. Information on the use of ascorbate supplements and high dose ascorbate injections was collected.

### 2.7. Statistical Analyses

Clinical and demographic characteristics between cohorts were compared with *t*-tests with Satterthwaite’s adjustment for unequal variance or Fisher’s exact test for continuous and categorical measurements, respectively. Categorical variables were compared with analysis of variance (ANOVA) followed by Tukey’s post-hoc tests and relationships between continuous variables were described by locally weighted regression smoothers and Pearson’s correlation coefficient. All continuous variables had no linear effect on plasma ascorbate (*p* > 0.022) so were appropriately categorized for regression analyses. Effect of demographics (age, gender, ethnicity), health (smoking, exercise, body mass index (BMI)), plasma intake and supplementation on plasma concentrations (continuous) were compared with simple linear regression. There was sufficient power to fit multiple linear regression models with 2-way interactions. Variables which did not improve fit were sequentially removed and the final model was graphically assessed for goodness of fit. Data were analyzed in R version 3.6.4 (R Foundation for Statistical Computing, Vienna, Austria), all *p* values are two-tailed, unpaired and with statistical significance set at *p* < 0.05.

## 3. Results

### 3.1. Characteristics of the Two Cohorts of Patients with Cancer

The pre-surgical cohort comprised of 99 patients with 81 deemed fasting (<45 mg ascorbate intake on day of blood draw), and the therapy cohort comprised of 100 patients receiving chemotherapy or immunotherapy with 69 fasting (Appendix A). Breast and colorectal cancer were the two most frequent cancer types in both cohorts (Appendix A). Most patients in the therapy cohort received chemotherapy (including adjuvant, neoadjuvant and palliative therapy), and 17 received immunotherapy (14 pembrolizumab and 3 nivolumab, all melanoma).

A comparison was made between fasting, non-fasting and all patients (combined fasting with non-fasting) in the two cohorts (Appendix A). As patients were recruited at their clinic visits, many had already eaten on the day of the blood test. Fasting status was therefore determined following an assessment of the ascorbate content of food consumed prior to the blood sampling, with an upper limit of 45 mg vitamin C consumed in the previous 6 h for the fasting group. Mean dietary ascorbate intake in patients from the pre-surgical cohort, according to 24-hour recall, was similar in fasting, non-fasting and all patients (61.0 ± 6.2, 67.3 ± 14.9 and 61.5 ± 5.7 mg ascorbate, respectively).

In patients from the therapy cohort, fasting patients had lower levels compared to non-fasting and all therapy patients (61.9 ± 6.7, 129.5 ± 43.6 and 82.8 ± 14.4 mg ascorbate, respectively). In the pre-surgical cohort, non-fasting patients had predominantly (82%) saturating plasma ascorbate levels (14/17), whereas only 55% of the non-fasting patients in the therapy cohort had saturating levels (17/31). Three-quarters of the pre-surgical cohort reported recent or regular supplement use (13/17), compared to 61% (19/31) of the treatment cohort. Although supplement use was common in both cohorts, high dose ascorbate infusions were rare (one in pre-surgical and four in treatment cohorts, Appendix A). Plasma ascorbate levels were higher in the pre-surgical cohort compared to therapy cohort in fasting, non-fasting and all patients (pre-surgical 57.2 ± 2.7, 104.8 ± 6.5 and 65.4 ± 3.1 µM and therapy 46.8 ± 3.2, 73.8 ± 5.5 and 55.1 ± 3.0 μM ascorbate, respectively, Appendix A). It is noteworthy that although ascorbate intake was generally higher in the therapy cohort compared to the pre-surgical cohort, plasma ascorbate levels were lower.

Recent ascorbate intake temporarily increases plasma concentrations [28,29], with plasma levels increasing by at least 20 µM from 1–6 h following intake of 200 mg ascorbate from a food source or supplement [28]. Hence, fasting measurements are a more reliable indicator of body status, and subsequent ascorbate data is presented only for those who had ascorbate intake below 45 mg on the day of blood draw—defined as fasting patients (Table 1).

Most fasting patients in both cohorts were female, non-Māori/non-Pacifica with a BMI >25 (Table 1). Patients on therapy were, on average, 5 years younger than those in the pre-surgical group (*p* = 0.017). Current smokers represented 13% of patients in the therapy cohort, compared to 5% in the pre-surgical cohort (*p* > 0.05), and approximately one quarter of patients in both cohorts reported doing less than one hour of exercise per week. As expected, most patients in the pre-surgical cohort had localized disease (TNM stage 1–3), whereas patients in the therapy cohort had a majority of disseminated disease or recurrence (*p* = 1.30 *×* 10^−7^). There were no other significant differences in patient characteristics or behavior between the two cohorts.

### 3.2. Ascorbate Intake versus Plasma Ascorbate Status of Fasting Patients with Cancer

Reported ascorbate intake was on average 60–62 mg/day and was similar between the two cohorts that met fasting criteria (Appendix A, Figure 1A). Similarly, intake divided into categories of below NZ RDI (<45 mg/day), between NZ and US RDI (45–90 mg/day), and above US RDI (>90 mg/day), did not differ between the two cohorts (Table 1). Almost 1/5 of patients reported regular supplement use (Table 1).

Mean fasting plasma ascorbate concentrations in the pre-surgical cohort were significantly higher than levels in the therapy cohort (*p* = 0.013, Figure 1B). Mean fasting plasma concentrations in the pre-surgical cohort were 57.2 ± 2.7 μM, compared to 46.8 ± 3.2 μM in the treatment cohort. Patients in the pre-surgical cohort who consumed above NZ RDI ascorbate had significantly higher plasma ascorbate concentrations than those consuming below (*p* < 0.01, Figure 1C). However, patients in the therapy cohort did not show this benefit, and had significantly lower plasma concentrations than patients in the pre-surgical cohort even when consuming above NZ RDI (*p* < 0.05, Figure 1C).

Fewer patients in the pre-surgical cohort were ascorbate deficient (<11 μM, 1 vs. 7) or were marginally deficient (11–23 μM, 7 vs. 8), compared to the therapy cohort (*p* = 0.027, Table 2). About one quarter of the pre-surgical cohort had inadequate plasma ascorbate levels (23–50 μM) compared to more than one third of the therapy cohort (*p* = 0.026). These data indicate that undergoing therapy is associated with odds of 2.4 of having inadequate (23–50 µM) plasma ascorbate, and 3.2 of having low (<23 µM) plasma ascorbate levels (Table 2).

A weak association was evident between reported ascorbate intake (24-hour recall) and measured fasting plasma concentrations in patients with cancer. The ‘smoother’ graph in the pre-surgical cohort showed a linear relationship for ascorbate intake below 90 mg (r = 0.42, *p* = 0.0007), which is lost at ascorbate intake above 90 mg (Figure 2). The therapy cohort showed a linear but less steep incline of plasma ascorbate with increasing intake up to ~60 mg, which again is lost at higher estimated ascorbate intake; neither are significant for the entire cohort (Figure 2).

### 3.3. Associations between Plasma Concentrations of Ascorbate and Patient Characteristics

Fasting plasma ascorbate levels were significantly lower in men compared to women in the pre-surgical cohort (*p* = 0.005), with a similar trend seen in the therapy cohort (*p* = 0.08) (Figure 3A,B). This was seen despite there being no difference in intake of ascorbate by gender (Figure 3C,D). When both cohorts were combined, women consuming above NZ RDI had higher plasma ascorbate levels than women who consumed lesser amounts, and also higher levels than men who consumed above NZ RDI (Figure 3E). Similarly, women consuming below NZ RDI had higher plasma ascorbate levels than men consuming below NZ RDI (Figure 3E).

Patients in the pre-surgical cohort doing less than 60 min exercise per week had significantly lower fasting plasma ascorbate concentrations than those doing more exercise each week (*p* = 0.004, Figure 4A). A similar trend was seen in patients from the therapy cohort, but this did not reach significance (*p* = 0.11, Figure 4B). Post-hoc tests on the pre-surgical cohort showed no significant difference between exercise at 60–150 min or >150 min (*p* = 0.96) so exercise was collapsed to two levels at 60 min for subsequent analyses (Figure 4C). In both cohorts combined, patients who consumed below NZ RDI and did less than 60 min exercise per week had significantly lower plasma ascorbate levels than those doing more exercise, regardless of their ascorbate intake (*p* < 0.001, Figure 4C).

### 3.4. Modeling of Associations with Plasma Ascorbate

The effect of patient characteristics on ascorbate status was further explored in all fasting patients combined using univariate analysis (Table 3). Male gender, low ascorbate intake (<45 mg/day), low exercise (<60 min/week) and being in the treatment cohort were associated with lower plasma ascorbate levels, whereas regular supplement use was associated with higher plasma ascorbate levels (all *p* < 0.05). Higher cancer stage was weakly associated with lower plasma ascorbate (*p* = 0.084), whereas age, BMI, smoking status, and ethnicity were not associated with ascorbate status (Table 3). Note, cancer stage and treatment were interdependent; most patients receiving therapy had high stage disease, and most patients in the pre-surgical cohort had lower stage disease (Table 1).

Multiple linear regression (Table 4) demonstrated that male gender was associated with lower plasma ascorbate. Daily ascorbate intake >45 mg was associated with higher plasma ascorbate, but this was significantly reduced in the cohort receiving treatment. Exercising more than 60 min per week was associated with a significant increase in plasma ascorbate, but only in those with ascorbate intake less than 45 mg. Together all predictors explained 26% of the variability (multiple R^2^), however age, BMI, ethnicity and smoking were not significant.

## 4. Discussion

Our study demonstrated that a significant proportion of patients with cancer had inadequate plasma ascorbate levels, consumed below NZ RDI for ascorbate (<45 mg/day) and exercised less than recommended (<60 min/week). Patients in the therapy cohort had significantly lower plasma ascorbate levels compared to those in the pre-surgical cohort. Men had lower ascorbate status than women, and patients who exercised for less than one hour per week had lower ascorbate levels than those doing more than one hour per week.

Average plasma ascorbate levels in patients with more advanced disease and who were receiving chemo- or immunotherapy were 10 μM lower than levels measured in the pre-surgical cohort. This discrepancy could not be explained by reduced ascorbate intake (e.g., due to treatment-induced nausea) in the therapy cohort. Indeed, our data showed that ascorbate intake was similar or higher in patients in the therapy cohort compared to those prior to surgery, yet plasma ascorbate levels were lower. This is an interesting observation and may be due to increased requirements of ascorbate in patients with advanced cancer. Only about half of non-fasting therapy patients had saturating plasma ascorbate levels compared to 82% of the non-fasting pre-surgical cohort. These combined data support the notion that patients currently undergoing cancer therapy are at increased risk of low ascorbate status.

Our analysis indicates a more complex relationship between ascorbate in the blood and dietary intake for cancer patients than has been reported in healthy individuals. For lower levels of intake, the data showed a linear increase in plasma concentrations with increasing intake, similar to what has been described in healthy volunteers [29], and levels appeared to (at best) plateau as intake reached the RDI recommended in Europe and the USA (90 mg/day). In fact, not many patients reached saturation levels. However, importantly, the patients with more advanced disease undergoing therapy appeared to require higher ascorbate intake to achieve similar plasma concentrations than those with localized disease prior to cancer surgery. Ascorbate supports many aspects of human health, including immune health and mood, and can support the health, quality of life and outcomes for cancer patients [30,31,32,33,34]. Independent studies are needed to confirm the relationship between treatment and ascorbate status observed in our study.

Our treatment cohort had 6/69 (8.7%) patients with ascorbate deficiency, compared to 1/81 (1.1%) in the pre-surgical cohort, and 9/369 (2.4%) in a cohort of healthy 50-year old’s [25]; with an additional 11.6% with marginal deficiency, a total of 1/5. Low plasma ascorbate levels have also been reported in patients receiving intensive chemotherapy [35]. This may have serious clinical implications, including poor wound healing, depression, poor response to therapy and risk of cancer death [30]. A previous study reported deficiency in 30% (15/50) of patients with advanced cancer and found that deficient patients had significantly shorter mean survival (29 versus 121 days, *p* = 0.001) [22]. Health can also be impaired in those patients with marginal ascorbate deficiency or inadequate levels. In a cohort of 598 kidney transplant recipients, low plasma ascorbate status was associated with risk of death from cancer [36]. Kidney transplant recipients are known to be at increased risk of several cancer types due to ongoing immune-suppression [37], and there is growing evidence of the importance of ascorbate in immune cell function [31].

We showed an association between low activity levels and low ascorbate levels in patients with cancer. Similar findings were reported for healthy elderly Japanese women (*n* = 655), where plasma ascorbate was correlated with handgrip strength, balance and walking speed [38]. Support was also recently published from the European Prospective Investigation into Cancer (EPIC)-Norfolk study [39]. This population-based cross-sectional study in England (*n* = 22,474 of largely healthy adults) showed an association between ascorbate deficiency and reduced reported physical activity [39]. Likewise, in cohorts of pre-diabetic and diabetic individuals (*n* = 89), plasma ascorbate and physical activity were related [40]. A small intervention trial in healthy men (*n* = 28) showed a slight improvement in physical activity with ascorbate supplementation [41]. These studies in healthy adults support our findings in patients with cancer, although it remains to be shown whether low activity levels are a reflection of poor health and diet, or whether low ascorbate, via its requirement as enzyme cofactor associated with energy metabolism and vitality [11], directly affects activity levels.

Male gender was associated with lower ascorbate in our study and others [39]. This does not appear to depend on ascorbate intake, and a higher requirement for ascorbate in males has previously been reported in healthy individuals [6,42]. Accordingly, the RDI for males is often higher for men than for women [43]. The reasons for increased requirement for males may indicate increased average body size, or an increased requirement for the vitamin, but this remains unclear.

We saw no associations between plasma ascorbate and age, BMI or smoking, which contrasts with reports from individuals without cancer [39,40]. The importance of these differences is not yet clear, but may be due to low numbers or may reflect the impact of cancer burden in our study.

The strength of our study is the reasonable size of the cohort which has statistical power to assess ascorbate in relationship to numerous health risk factors (only one other study contained more than 150 patients with cancer [44]). Other strengths are robust measurements (HPLC) of ascorbate, and sampling at different stages of the cancer continuum. Limitations are that fasting ascorbate data was only available for 150 of the 199 patients, dietary intake was estimated from a single survey time point, and exercise was assessed from a simple survey. Additional factors that potentially affect ascorbate status, such as systemic inflammation, were not assessed.

## 5. Conclusions

Our data demonstrates that plasma ascorbate levels in cancer patients are driven by more than ascorbate intake or any single health risk factor. Modeling showed male gender, sedentary behavior and current cancer treatment to be predictive of risk of low ascorbate status. Based on the present study findings, it appears that patients with advanced cancer currently undergoing chemo- or immunotherapy may be at particular risk of ascorbate depletion associated with increased requirements, and not necessarily due to reduced intake. As ascorbate deficiency is a severe health risk, this information is valuable for clinicians when advising their patients on health and dietary choices, including consideration of supplement use. It is of note that the vast majority of people with cancer will never have their plasma ascorbate measured, and this report may prompt increased vigilance of ascorbate status in cancer patients.

## Figures and Tables

**Figure 1 nutrients-12-02338-f001:**
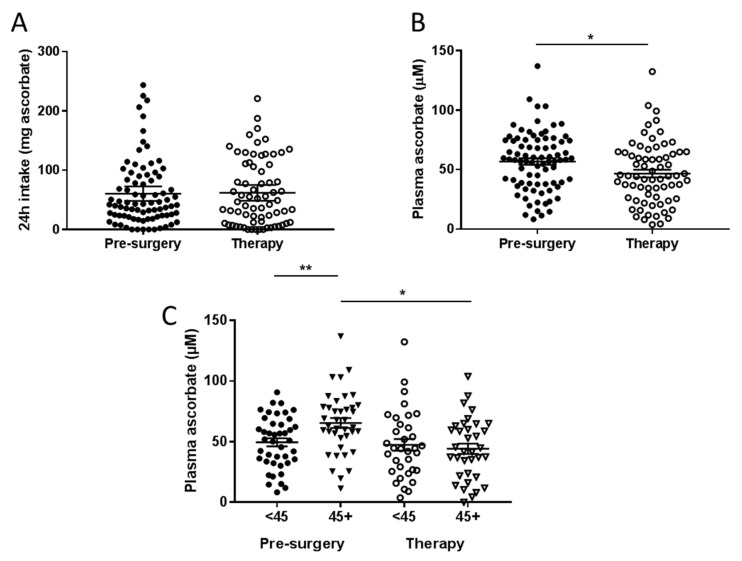
Ascorbate intake (**A**) and plasma ascorbate concentrations (**B**) in fasting patients from the pre-surgical and therapy cohorts. (**C**) Plasma ascorbate levels according to reported intake dicotemized at NZ recommended daily intake (RDI) of 45 mg/day. Pre-surgical cohort *n* = 81; therapy cohort *n* = 69. * *p* < 0.05, ** *p* < 0.01, unpaired *t*-test; mean ± standard error of the mean (SEM).

**Figure 2 nutrients-12-02338-f002:**
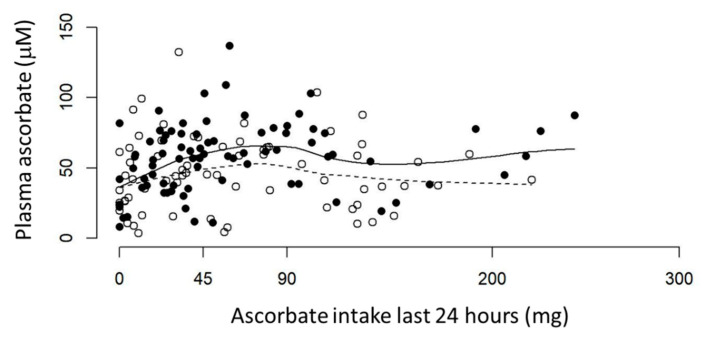
Ascorbate intake vs. plasma ascorbate levels in fasting patients with cancer. Pre-surgical (•) and therapy (°) cohorts are shown with lines locally weighted ‘smooths’ for each group. Solid line shows pre-surgical cohort, dotted line shows therapy cohort. Plasma concentrations are from fasting patients (<45 mg ascorbate on day of blood draw) and ascorbate intake is from 24-hour dietary recall. Pre-surgical cohort *n* = 81, therapy cohort *n* = 69.

**Figure 3 nutrients-12-02338-f003:**
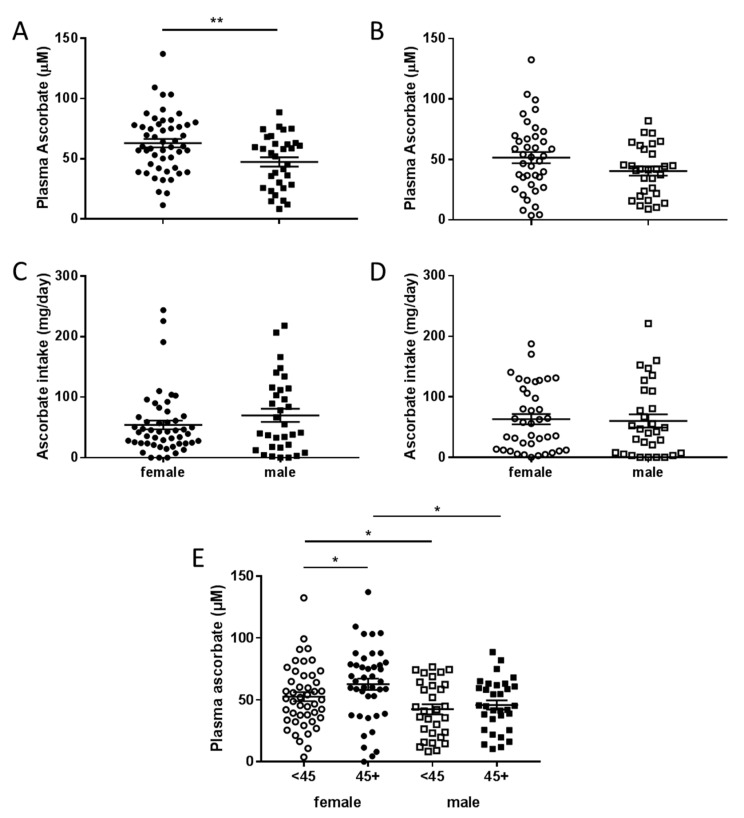
Plasma ascorbate levels and intake according to gender in fasting patients. Plasma data is compared between female and male patients from the pre-surgical (**A**) and therapy cohorts (**B**), intake data (24-hour dietary recall) from the pre-surgical (**C**) and therapy cohorts (**D**). (**E**) Association of plasma ascorbate and gender with intake dicotemized at NZ RDI of 45 mg/day. Unpaired *t*-test, * *p* < 0.05, ** *p* < 0.01, pre-surgical cohort *n* = 81, therapy cohort *n* = 69, mean ± SEM.

**Figure 4 nutrients-12-02338-f004:**
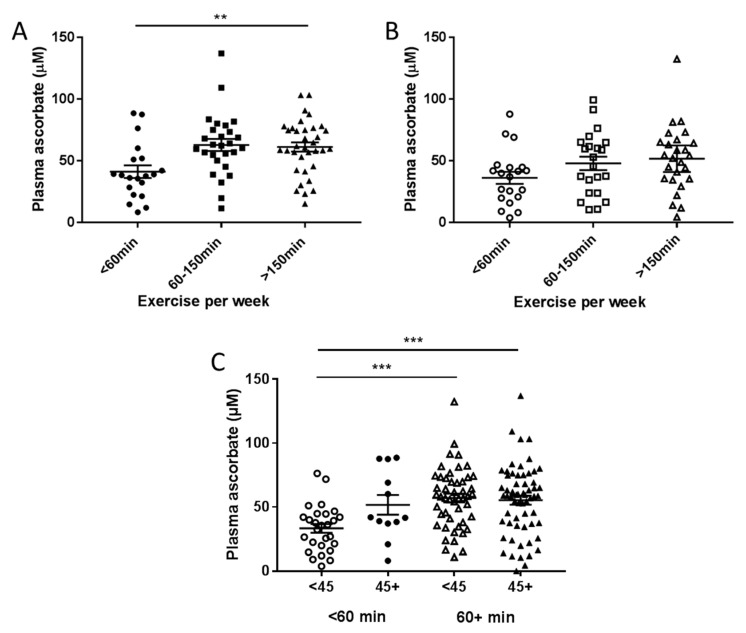
Fasting plasma ascorbate levels according to reported exercise levels in patients from the pre-surgical (**A**) and treatment cohorts (**B**). (**C**) Association of plasma ascorbate and exercise with intake dicotemized at NZ RDI of 45 mg/day and exercise dicotemized at 60 min/week. Pre-surgical cohort *n* = 81, therapy cohort *n* = 69. ** *p* = 0.004 one-way analysis of variance (ANOVA), *** *p* < 0.001 unpaired *t*-test, mean ± SEM.

**Table 1 nutrients-12-02338-t001:** Comparison of characteristics of the two cohorts of fasting patients with cancer.

		Pre-Surgical Cohort *n* = 81 (%)	Therapy Cohort *n* = 69 (%)	Effect Size [95% CI]	*p* Value
**Age (years)**	Mean (±SD)	63.88 (±12.08)	58.71 (±13.76)	5.17 [0.95,9.38]	**0.017**
**Gender**	Female	49 (60)	39 (57)		
	Male	32 (40)	30 (43)	1.18 [0.61,2.26]	0.740
**Ethnicity**	European	72 (89)	63 (91)		
	Māori/Pacifica	9 (11)	6 (9)	0.76 [0.26,2.26]	0.786
**BMI (kg/m^2^)**	Mean (±SD)	30.50 (±7.17)	28.62 (±7.13)	1.88 [−0.45,4.20]	0.112
**Smoking**	never	42 (52)	35 (51)		
	ex	35 (43)	25 (36)	0.86 [0.43,1.69]	0.730
	current	4 (5)	9 (13)	2.70 [0.77,9.52]	0.140
**Exercise (min/week)**	>150	35 (43)	26 (38)		
	60–150	27 (33)	22 (32)	1.10 [0.51,2.34]	0.848
	<60	19 (23)	20 (29)	1.42 [0.63,3.18]	0.418
**Stage**	TNM 1–3	67 (83)	30 (43)		
	TNM 4, recurrence	12 (15)	39 (57)	7.26 [3.34,15.79]	**1.30 × 10^−7^**
**Ascorbate intake (mg/day)**	<45	42 (52)	34 (49)		
	45–90	19 (23)	15 (22)	0.98 [0.43,2.20]	1.000
	≥90	20 (25)	20 (29)	1.24 [0.57,2.66]	0.696
**Supplement**	No	66 (81)	56 (81)		
	Yes	15 (19)	13 (19)	1.02 [0.45,2.33]	1.000

Age and body mass index (BMI) show mean (±standard deviation) and are compared with *t*-tests with Satterthwaites adjustment for unequal variance. Effect is difference [95% Confidence Interval]. Categories show counts (%) and are compared with Fisher’s exact test with effect shown as odds ratio [95% confidence interval (CI)]. There is 1 participant who received therapy with no exercise record, and 2 pre-surgical patients with no stage information. TNM stage, tumor node, metastasis stage. Significant *p*-values are shown in bold.

**Table 2 nutrients-12-02338-t002:** Comparison of fasting cancer patient cohorts according to categories of plasma ascorbate.

Plasma Ascorbate	Pre-Surgical Cohort	Therapy Cohort	OR [95% CI]	*p*-Value
	*n* = 81 (%)	*n* = 69 (%)		
>50 μM	53 (65.4)	29 (42.0)	1	
23–50 μM	20 (24.7)	26 (37.7)	2.38 [1.14, 4.97]	**0.026**
<23 μM	8 (9.9)	14 (20.3)	3.20 [1.20, 8.52]	**0.027**

OR odds ratio, CI confidence interval, *p* value from Fisher exact test relative to >50µM plasma ascorbate level. Significant *p*-values are shown in bold.

**Table 3 nutrients-12-02338-t003:** Univariate effects of patient characteristics on plasma ascorbate.

		Estimate	95% CI	*p*-Value
**Age**	50–70	3.39	[−7.41,14.19]	0.415
	70+	−3.21	[−15.49,9.07]	
**Gender**	Male	−13.97	[−22.14,−5.79]	**0.001**
**BMI**	30–40	2.63	[−6.68,11.93]	0.719
	<18.5 or >40	−3.57	[−18.71,11.56]	
**Smoking**	ex	−5.69	[−14.46,3.07]	0.272
	current	−9.90	[−25.16,5.37]	
**Ethnicity**	Māori/Pacifica	3.19	[−10.73,17.11]	0.651
**Exercise**	60–150	−1.09	[−10.37,8.20]	**0.001**
	<60	−18.24	[−28.15,−8.32]	
**Ascorbate Intake**	45–90	9.76	[−1.96,21.47]	**0.049**
	<45	−3.10	[−13.01,6.81]	
**Supplementation**	yes	12.32	[1.79,22.86]	**0.022**
**Tumor Stage**	TNM 1–3	−7.16	[−18.29,3.97]	0.084
	TNM 4, recurrent	−10.42	[−19.83,−1.00]	
**Cohort**	Treatment	−10.43	[−18.64,−2.22]	**0.013**

Estimates, 95% confidence intervals and *p* values from univariate linear regression on plasma ascorbate. Combined cohorts of fasting patients *n* = 150, age (years) vs. <50, BMI, body mass index (kg/m2) vs. 18.5–30, ethnicity Māori/Pacifica vs. other, exercise (min/week) vs. >150, ascorbate intake (mg/day) vs. >90, tumor stage TNM (tumor node metastasis) local vs. metastatic or recurrent disease. Significant *p*-values are shown in bold.

**Table 4 nutrients-12-02338-t004:** Multiple linear regression of plasma ascorbate in fasting patients with cancer.

Predictor	Level	Effect	95% CI	*p*-Value
**Gender**	Male	−12.31	[−19.73,−4.89]	**0.0013**
**Ascorbate Intake**	>45 mg	29.74	[12.47,47.01]	**0.0009**
**Exercise**	>60 min	22.08	[11.40,32.77]	**0.0001**
**Ascorbate Intake × Exercise**	>45 mg and >60 min	−19.93	[−37.62,−2.25]	**0.0275**
**Cohort**	Therapy	0.02	[−10.24,10.28]	0.9968
**Ascorbate Intake × Cohort**	>45 mg and Therapy	−20.83	[−35.47,−6.18]	**0.0056**

Linear effect, 95% confidence interval and *p*-value from linear regression of plasma ascorbate on gender, ascorbate intake, exercise by ascorbate intake and cohort by ascorbate intake on combined cohorts of fasting patients *n* = 150. Significant *p*-values are shown in bold.

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
