# Peer review of "Low Vitamin C Status in Patients with Cancer Is Associated with Patient and Tumor Characteristics"

_nutrients, 2020, doi:10.3390/nu12082338_

Round 1

Reviewer 1 Report

In this report, the authors aimed to accurately measure ascorbate concentrations in plasma from cancer patients and determine associations with patient or tumor characteristics. As a result, a significant number of cancer patients had inadequate plasma ascorbate concentrations. In addition, low plasma status was more prevalent in patients undergoing chemotherapy or immunotherapy than in patients undergoing cancer surgery. Ascorbate status was higher in women than in men, and exercising patients had higher levels than sedentary patients. The authors concluded that increased monitoring of ascorbate status may be required in cancer patients. The present study is well performed comprehensive analysis relating to ascorbic status in the setting of malignancy, although I have some minor comments to discuss.

Minor comments

  1. Introduction line 59–61: The authors stated that previous studies on the ascorbate status of cancer patients involved small sample sizes. Meanwhile, the present study also recruited relatively small number of fasting patients with cancer (n = 150). The authors should mention this in the limitation section.
  2. Results line 145–149: The authors concluded that the dietary ascorbate intake in fasting patients in the therapy cohort (61.9 ± 6.7 mg) was lower compared to non-fasting and all therapy patients. However, this was rather similar compared to fasting, non-fasting, and all patients in the pre-surgical cohort (61.0 ± 6.2, 67.3 ± 14.9 and 61.5 ± 5.7 mg ascorbate, respectively). It would be better to emphasize that the intake of ascorbate in cancer patients undergoing chemotherapy or immunotherapy was actually increased (129.5 ± 43.6 mg), although plasma ascorbate levels were not higher in the therapy cohort (73.8 ± 5.5 μM) compared to pre-surgical cohort (104.8 ± 6.5 μM). This is an interesting finding and may be due to increased requirements of ascorbate in patients with advanced cancer (which was indeed more prevalent in the therapy cohort).
  3. Introduction line 69–70: Therefore, based on the present study findings, therapy patients may be particular risk of ascorbate depletion due to increased requirements, but not from reduced intake.
  4. There are several grammar errors which may be improved by revision.

Author Response

Please see attached our detailed response

Reviewer 2 Report

I think ref 5 is not optimal for judging clinical results, since it is about evidence on the mechanism..

I would favour an other paper (van Gorkom Nutrients 2019 would be an option).

This paper stresses a lot on statistical significance, but little on clinical relevance.

I.e. the level of inadequate ascorbis acid levels (what is the exact meaning of this term?) goes to 50MicroM.   the difference between 57 and 46 (pre-surgical cohort) and treatment cohort (46.8) seems limited in relation to the 50 as inadeqaute. I feel the authors should say more (in the discussion) on the interpretation of the differences observed (at several moments).  Looking at the figures at a glance there seems to be more overlap between the groups than differences, despite the stars.

maybe my statistical approach is different but can the authors explain how in table  2 the OR for 23-50micromol between the two groups is 2,37 where the absolute values are 24.5% versus 37.7 %.  same for <23 microM. 9.9 % versus 20.3% does give an OR of 3.198 (three points after the 3 is not realistic anyway. I would guess about 2.

I feel the conslusion is far from acurate. I am not so sure about the severe health risk of low vitamine C status in patients with cancer (if the authors refer to ref 29).  For cancer patients the association between Vit C and death due to cancer is borderline significant and incosistent among different tumor types in this study.  Moreover this  reference and the data presented here do not support monitoring of ascorbate status of patients unless we have intervention trials.

Finally can the authors say anything on the relevance of plasma values for vitamin C  status.  Or should we measure levels in tumors and or leukocytes?  like FE in blood does not correlate to  iron status.

Author Response

(The authors gave the same response as above.)

Reviewer 3 Report

This original research provides a well-defined question and is of relevance to care nutritional management of cancer patients. The results of the study show that vitamin C status is poor in cancer patients and correlations to certain factors are established. The write up does not provide a full description of influencing facts on vitamin C status (but this has only recently been fully reported in the literature). Not all of the potential factors influencing vitamin C levels are discussed in the text. Key factors here include assessment of systemic inflammation (from recent infections, surgery, advanced disease). Ideally an assessment of CRP would have been included, which is known to correlate with vitamin C levels, if not available then this should be included in the limitations of the study. A recent article by Carr & Rowe looks at factors effecting deficiency in more detail (Carr AC, Rowe S. Factors Affecting Vitamin C Status and Prevalence of Deficiency: A Global Health Perspective. Nutrients. 2020;12(7):E1963. Published 2020 Jul 1. doi:10.3390/nu12071963).

The study identifies that serum total vitamin C levels are often low in cancer patients. As most centres don’t have research facilities for vitamin C testing and it is currently costly, I think alternative interventions should be discussed in the text. Do these findings not highlight the need for all cancer patients to receive adequate nutrition, not just to meet the measly 45 mg minimum requirements in current guidelines?

Further emphasis on the findings for vitamin C status from other studies could be highlighted. Particularly in the background section.

This study was underpowered as a result of the inclusion/exclusion criteria. Whilst this is briefly mentioned I think it deserves further emphasis in the write-up, it’s a waste of resources and arguably unethical to enrol patients in a study where their results will be excluded ( for non-fasting patients and those on supplements). This is a real shame when the research team have followed such a meticulous approach to the analysis and handling of samples.

I think that the findings of vitamin C status being low amongst cancer patients is  of interest to all clinicians involved in the care of cancer patients. The associations with specific patients’ groups are of some interest and more specific to specialists.

I think the authors need to re-evaluate the implications of the research and build on their recommendations. Should the recommended intake be increased? Should supplements be offer to those with poor intake? Should diet be routinely assessed?

A few other minor points are:

  • The title may benefit from a more generalised name referring to "low Vitamin C status" rather than insufficiency as this is poorly defined in the literature on vitamin C.
  • Presumably the decision to exclude patients from the final analysis was made as a change in protocol? If so this should be clearly described in the paper. Was this performed following the initial analysis? If so then this needs to be acknowledged in the text.
  • Table 2 would benefit from a line on frank deficiency as well as hypovitaminosis.
  • Could elaborate on the effects of improving status on immune function, rates of infections and mood.

Author Response

(The authors gave the same response as above.)
